# Fabrication and Characterization of a Poly(3,4-ethylenedioxythiophene)@Tungsten Trioxide–Graphene Oxide Hybrid Electrode Nanocomposite for Supercapacitor Applications

**DOI:** 10.3390/nano13192664

**Published:** 2023-09-28

**Authors:** Cherifa Hakima Memou, Mohamed Amine Bekhti, Mohamed Kiari, Abdelghani Benyoucef, Magbool Alelyani, Mohammed S. Alqahtani, Abdulaziz A. Alshihri, Youssef Bakkour

**Affiliations:** 1Laboratory of Physical and Macromolecular Organic Chemistry, Faculty of Exact Sciences, Djillali Liabes University, Sidi Bel Abbes 22000, Algeria; 2LCOMM Laboratory, University of Mustapha Stambouli Mascara, Mascara 29000, Algeria; 3Department of Chemical and Physical Sciences, Materials Institute, University of Alicante (UA), 03080 Alicante, Spain; 4LSTE Laboratory, University of Mustapha Stambouli Mascara, Mascara 29000, Algeria; 5Department of Radiological Sciences, College of Applied Medical Science, King Khalid University, Abha 61421, Saudi Arabia

**Keywords:** poly(3,4-ethylenedioxythiophene), tungsten trioxide, graphene oxide, ternary nanocomposite, supercapacitor

## Abstract

With the rapid development of nanotechnology, the study of nanocomposites as electrode materials has significantly enhanced the scope of research towards energy storage applications. Exploring electrode materials with superior electrochemical properties is still a challenge for high-performance supercapacitors. In the present research article, we prepared a novel nanocomposite of tungsten trioxide nanoparticles grown over supported graphene oxide sheets and embedded with a poly(3,4-ethylenedioxythiophene) matrix to maximize its electrical double layer capacitance. The extensive characterization shows that the poly(3,4-ethylenedioxythiophene) matrix was homogeneously dispersed throughout the surface of the tungsten trioxide–graphene oxide. The poly(3,4-ethylenedioxythiophene)@tungsten trioxide–graphene oxide exhibits a higher specific capacitance of 478.3 F·g^−1^ at 10 mV·s^−1^ as compared to tungsten trioxide–graphene oxide (345.3 F·g^−1^). The retention capacity of 92.1% up to 5000 cycles at 0.1 A·g^−1^ shows that this ternary nanocomposite electrode also exhibits good cycling stability. The poly(3,4-ethylenedioxythiophene)@tungsten trioxide–graphene oxide energy density and power densities are observed to be 54.2 Wh·kg^−1^ and 971 W·kg^−1^. The poly(3,4-ethylenedioxythiophene)@tungsten trioxide–graphene oxide has been shown to be a superior anode material in supercapacitors because of the synergistic interaction of the poly(3,4-ethylenedioxythiophene) matrix and the tungsten trioxide–graphene oxide surface. These advantages reveal that the poly(3,4-ethylenedioxythiophene)@tungsten trioxide–graphene oxide electrode can be a promising electroactive material for supercapacitor applications.

## 1. Introduction

Supercapacitors (SCs) have received enormous attention due to their excellent advantages, including long cycle life, high power density, good safety, rapid charging/discharging rate, and lower maintenance cost [1,2,3]. Although they have been widely applied in various fields, including electric vehicles, pulse power systems and portable devices, the traditional SCs have some obvious shortcomings, such as a large volume, heavy weight and are difficult to deform, and are far from the requirements for the rapid development of wearable electronics [2,4]. Therefore, there is a great challenge to overcome these major drawbacks in the design and synthesis of flexible high-performance SCs and their corresponding electrode materials [2]. Moreover, some of the electrode materials studied for effective storage of charge, include the transition metals, carbonaceous material, and conductive polymeric materials. For use in SC applications, carbonaceous electrode materials such as mesoporous carbon, graphene oxide (GO), reduced graphene oxide, carbon nanofiber, activated carbon, and carbon nanotubes exhibit outstanding rate capability, good reversibility, and excellent stability [2,5,6,7]. On the other hand, GO is widely used for charge storage applications in SCs. This is due to their outstanding properties such as high electrical and thermal conductivity, vast surface area and chemical stability. Furthermore, GO is a flexible lamellar material that has a wide range of functional groups on both basal planes and edges. As a result, it can be easily exfoliated and functionalized to form homogeneous suspensions in both water and organic solvents, providing more possibilities for the synthesis of graphene-based materials. The existence of oxygen functional groups and aromatic sp^2^ domains allows GO to participate in a wide range of bonding interactions [5,8]. Due to these positive properties with agglomerated and stacked structures, it was found that GO had a specific capacitance (Csp) of about 150 F·g^−1^ [9]. The capacitive performance can be improved by introducing metal oxides (MOs) in between GO sheets in order to reduce the agglomeration and stacking. Due to their high surface activity, exceptional electrochemical characteristics, improved, and changeable oxidation states, MOs have recently attracted a great deal of research attention in SCs [1,10,11]. It was also reported that the MOs/GO hybrid materials deliver high Csp and high robust cycling stability. For example, Shi et al. [12] reported fast facile synthesis of SnO_2_/graphene composite assisted by microwave as the anode material for lithium-ion batteries and it delivered a Csp of 112.0 F·g^−1^ with a capacitance retention as high as 73.2% after continuous 2000 cycles. Gao et al. [13] investigated the design and preparation of a graphene/Fe_2_O_3_ nanocomposite as a negative material for a supercapacitor, and obtained a Csp that was 378.7 F·g^−1^ at a current density of 1.5 A·g^−1^, and a Csp retention of 88.76% after 3000 cycles. Li et al. [14] studied a three-dimensional hierarchical graphene/TiO_2_ composite as an electrode for a supercapacitor which had a Csp value of 235.6 F·g^−1^ at 0.5 A·g^−1^ with a cycling stability of 90% after 500 cycles. Sheikhzadeh, Gu et al. [15] reported a nanocomposite foam layer of CuO/graphene oxide for a high performance supercapacitor with a Csp of 238.3 F·g^−1^. Sahoo et al. [16] reported a vanadium pentaoxide-doped waste plastic-derived graphene nanocomposite for supercapacitors with a Csp of 58.15 F·g^−1^ at 1.0 A·g^−1^. Qiu et al. [17] also reported scalable sonochemical synthesis of petal-like MnO_2_/graphene hierarchical composites for high-performance supercapacitors which had a Csp of 187.2 F·g^−1^ at 0.5 A·g^−1^.

Tungsten oxide (WO_3_) is an electrochemically sustainable MO of type n with implementation in different sectors. Its 2D layered structure, similar to GO sheets, permits high reversibility in the intercalation of ions, high electronic conductivity and large current capacity [5]. In addition to their superior effective applications in energy storage devices, photocatalysis, electrochromics, gas sensing and field-mission devices, nanostructured WO_3_ materials have gained growing interest [18,19]. This is due to their low charge movement resistance and significant surface area [19,20]. Moreover, WO_3_/GO materials exhibit advantageous electrochemical redox properties, high specific capacities, and reactivities of ions [19]. Different investigations have been carried out by researchers. The experimental results have shown that WO_3_/GO hybrid materials are potential candidates for SC applications due to the delivery of higher Csp values of 143.6 F·g^−1^, as compared to pure WO_3_ (32.4 F·g^−1^) at 0.1 A·g^−1^ [21].

To enhance the capacity of SCs, conducting polymers (CP) such as polyaniline (PANI), polypyrrole (PPy), and poly(3,4-ethylenedioxythiophene) (PEDOT) have offered an efficient solution [22,23]. PEDOT, owing to its facile availability, exceptional stability, processability, easy oxidation potential, short band gap, and thermal stability make it perfect for various applications such as energy storage [22,23,24]. However, most of the advantages of PEDOT-based SCs comes from vapor deposition because this provides the exceptional inherent conductivity suitable for SC electrodes. Consequently, exploring the synthesis approach of novel PEDOT electrodes is a useful method and a technique to effectively improve cycle stability and electrochemical performance. The in situ polymerization of EDOT with GO is possible; however, the bulk structures with low pores limit the access of electrolyte ions to the surface resulting in a deprived performance. To fully utilize the substrate as an electrical double-layer capacitor contributor and MO and a CP shield as a pseudocapacitor active material, it is important to grow the structures with structures suitable for ion penetration [25]. As recently reported, PEDOT, GO and nanocomposites of WO_3_ and conducting polymers can effectively improve the power density of supercapacitors [26,27,28,29,30,31,32]. However, the aggregation of nanomaterials easily occurs during the construction of 3D nanostructures when using the abovementioned materials, leading to a decrease in the specific surface area [33]. In the literature, studies have been found on the contribution of CP to GO with the presence of metal oxide [1,2,3,11,12,13,14,15]. Haldar et al. [1] produced via a chemical polymerization method, metal oxide/PANI/graphene composites using different metal oxides (ZrO_2_, WO_3_ and V_2_O_5_) and investigated its electrochemical properties. Bejjanki et al. [34] synthetized a SnO_2_/RGO@PANi ternary composite via chemical oxidation polymerization for supercapacitors. According to their results, they observed that adding metal oxide to the GO in the presence of CP matrix greatly affects the supercapacitor’s properties and increases the specific capacitance value. These results inspired our study.

In this work, we report a simple method to prepare PEDOT@WO_3_–GO ternary composite, starting with WO_3_–GO, which was considered a primary material, where WO_3_ was applied in situ on GO sheets using a simple chemical synthesis method. Secondly, polymer matrix was followed via chemical oxidation polymerization of EDOT. The PEDOT@WO_3_–GO electrode material so obtained has excellent electrochemical performance and a good specific capacitance of 478.3 F·g^–1^ and a maximum energy density of 54.2 Wh·kg^–1^ with a power density of 971 W·kg^–1^. This is attributed to a larger effective surface area of the fabricated materials. These findings indicate that the PEDOT@WO_3_–GO nanocomposites could be promising electrode materials for high-performance supercapacitor applications, which has not been previously reported.

## 2. Materials and Methods

### 2.1. Materials

Graphite powder (Superior Graphite Co., Ltd., Chicago, IL, USA. 99.9%), 3,4-ethylenedioxythiophene (EDOT; Sigma Aldrich, Madrid, Spain. ≥99.5%), tungsten trioxide (WO_3_; Merck, Shanghai, China. ≥99%), ammonium persulfate (APS; Merck, Riga, Lithuania. ≥98%), polyvinylidene fluoride (PVDF); ammonia solution (NH_4_OH; Merck, Riga, Lithuania. 25%), N-methylpyrrolidone (NMP; Merck), carbon black (CB, Superior Graphite Co., Ltd., Chicago, IL, USA) as conductive additive. Sodium hydroxide (NaOH; Merck, Riga, Lithuania. 37%), sulfuric acid (H_2_SO_4_; Merck, Riga, Lithuania. 90%), hydrogen peroxide (H_2_O_2_; Merck, 70%), sodium nitrate (NaNO_3_; Merck), potassium hydroxide (KOH; Merck, Riga, Lithuania), potassium permanganate (KMnO_4_; Merck, Riga, Lithuania), ethanol (C_2_H_5_OH; Merck, Riga, Lithuania. 96%), filter paper, and deionized water (DIW). A commercial grade stainless steel (SS) foil (thickness: 0.2 mm) was used as the substrate for electrode preparation.

### 2.2. Measurements

XRD patterns were measured by an X-ray diffractometer (CCDApex Bruker. Madison, WI, USA). TEM images were collected on an Hitachi H7500 (Tokyo, Japan) electron microscope. Surface area and pore volume analyses were performed on an iQinstrument Autosorb analyzer (Madrid, Spain). Prior to measurements, the samples were degassed at 120 °C under high vacuum overnight. The Brunauer–Emmett–Teller (BET) method was used to calculate the specific surface area of the materials. The pore size distribution was derived from Barret–Joyner–Halenda (BJH) method. FTIR spectra were recorded on a Bruker Alpha (Karlsruhe, Germany) spectrophotometer. The chemical composition of the materials was obtained via an X-ray photoelectron spectrometer (XPS) (AVG-Microtech-Multilab, 3000 electron, Tokyo, Japan). Thermogravimetric analysis (TGA) was used to check the thermal stability with a Hitachi (STA7200; Tokyo, Japan) instrument in an N_2_ atmosphere at a heat rate of 20 °C in the temperature range 0 °C to 900 °C. UV–Vis spectra were registered using a Hitachi spectrophotometer (U3000; Tokyo, Japan).

### 2.3. Synthesis of Graphene Oxide (GO)

Graphene oxide (GO) was prepared from graphite powder (GP) using a modified Hummer’s method [32]. GP (5 g) was first added to concentrated H_2_SO_4_ (115 mL) and NaNO_3_ (2.5 g) and stirred for 1 h with a magnetic stirrer, to which KMnO_4_ (15 g) was then slowly added and mixed for 1 h in an oil bath at 40 °C. Finally, to terminate the reaction, 10 mL of 30% H_2_O_2_ was added to the suspension. Then, the filtered material was washed until it reached a neutral pH. The washed product (GO) was dried for 24 h at 80 °C.

### 2.4. Preparation of WO_3_–GO

GO (100 mg) was added to 15 mL DIW and dispersed by ultrasound for 30 min to obtain a well-dispersed, negatively charged GO solution. Separately, 2.0 g of WO_3_ nanoparticles were ultrasonically dispersed in 20 mL NaOH solution (pH 9.5) for 30 min. Then, the two suspensions were mixed, and the sonication time was extended to 1 h with mild magnetic stirring at 50 °C to fulfill the electrostatic self-assembly process. Finally, the sediment solid was collected and washed and completely dried and annealed for 1 h at 300 °C to obtain the target sample of WO_3_–GO.

### 2.5. Fabrication of PEDOT@WO_3_–GO

1.0 mL EDOT was added to 25 mL 1M HCl by magnetic stirring. Then, the synthesized WO_3_–GO (1.0 g) discussed above, was added to it and ultrasonicated 30 min to disperse it properly. The temperature of the dispersion was reduced below 5 °C using an ice bath. Separately, 25 mL of HCl (1 M) solution was used to dissolve 2.5 g of APS. This solution was added dropwise to the dispersate under constant stirring for 6 h for the completion of the polymerization reaction and obtained a precipitate of green color. Then, the residue was filtered and washed with HCl, ethanol and DIW, and dried at 60 °C in an oven for 6 h. The obtained dry powder (PEDOT@WO_3_–GO) was collected and stored in a desiccator.

### 2.6. Electrochemical Studies

The electrochemical performance of the prepared electrodes was determined by a cyclic voltammetry (CV) technique. The CV was used in a three-electrode configuration, a platinum wire was used as a counter electrode (CE), a reversible hydrogen electrode (RHE) served as the reference electrode (RE) and the electrode of prepared material was used as the working electrode (WE). The electrochemical properties were evaluated at a fixed potential range of −0.1 V to +1.0 V in 3 M KOH as electrolyte at room temperature [22,23,24]. To prepare WE, active material of 70 wt%, CB of 15 wt% and PVDF of 15 wt%, were mixed in acetone and stirred at 60 °C until a homogeneous ink was obtained. Subsequently, the ink was drop-casted on a stainless steel (SS) plate (thickness: 1 μm) and dried at 60 °C overnight.

The specific capacitance Csp (F·g^−1^) from the electrode setup was calculated from CV and GCD by Equation (1):(1)$$\mathrm{Csp}=\frac{I\Delta t}{m}$$

The power density (E) and energy density (P) were calculated using galvanostatic discharge behavior via Equations (2) and (3):(2)$$E=\frac{1}{2}\mathrm{Csp}{\left(\Delta V\right)}^{2}$$
(3)$$P=\frac{E}{t}$$
where Csp denotes specific capacitance (F·g^−1^), V refers to the potential window (V), *m* is the mass of the active material (g), *I* is the discharge current density (A·g^−1^) and t (s) is the discharge time.

## 3. Results

### 3.1. Structural Determination

The synthesized WO_3_, WO_3_–GO, PEDOT and PEDOT@WO_3_–GO were characterized by using FTIR which is shown in Figure 1a. It can be observed that the WO_3_ nanoparticles give absorption bands at the positions of 615 cm^−1^, 726 cm^−1^ and 833 cm^−1^ and are attributed to the stretching and bending vibrations for O–W–O and W–O–W in WO_3_ [25,26]. The peak at 964 cm^−1^ is associated with the W=O stretching vibration and the peak at 1570 cm^−1^ is attributed to the hydroxyl group in W–OH. The spectra also show the renowned bands at 1670 cm^−1^ and 3423 cm^−1^, which are ascribed to H–O stretching and bending vibrational modes of free or absorbed water. Moreover, the results of the FTIR spectrum of WO_3_–GO confirms the existence of both components WO_3_ and GO in the nanocomposite; besides the occurrence of a shift towards higher frequency, new bands at 1055 cm^−1^, 1435 cm^−1^ and 1611 cm^−1^ of the GO spectrum appear and were due to C–O stretching, C–H bending and C=C stretching, respectively. In addition, the FTIR spectrum of PEDOT shows bands at 1582 cm^−1^, 1487 cm^−1^, 1373 cm^−1^ and 1208 cm^−1^ which are mainly due to C=C and C–C stretching of the quinoid structure of the thiophene rings. The band at 1151 cm^−1^ is due to C–O–C bond stretching in the ethylene dioxide units, while the bands at 932 cm^−1^ and 652 cm^−1^ are attributed to the C–S stretching mode. Furthermore, in the PEDOT@WO_3_–GO sample, the WO_3_–GO bands also appear with all spectra typical of bands for PEDOT. Compared with pure PEDOT, the band intensity of the PEDOT in the ternary nanocomposite was significantly weakened and broadened owing to the strong bonds between the PEDOT matrix and WO_3_–GO. The characteristic bands of the polymer chain tended to migrate slightly to a higher position. The formation of PEDOT@WO_3_–GO was confirmed.

The XRD patterns of materials are depicted in Figure 1b. A good crystalline nature of WO_3_ nanoparticles and also of WO_3_–GO has been evidenced by the intense diffraction peaks, but in PEDOT@WO_3_–GO the peaks are shifted from their respective standard positions and the intensity of peaks is reduced, which is due to the existence of the PEDOT matrix. The diffraction peaks in the XRD pattern of WO_3_ nanoparticles ascertained at 2*θ* = 28.13°, 23.47°, 24.33°, 26.63°, 28.95°, 33.22°, 34.16°, 41.85°, 47.24° and 49.97° can be readily indexed as (002), (020), (200), (120), (112), (022), (202), (222), (004) and (400) to lattice planes of WO_3_ monoparticles, respectively (JCPDS—00-044-0141). Moreover, the XRD diffraction pattern of WO_3_–GO reveals the characteristic peaks which belong to the WO_3_ nanoparticles with two new diffraction peaks at 2*θ* = 10.43° and 47.74° which are characteristic peaks of GO [26]. This result confirms the formation of GO. Since there were no peaks in the XRD pattern of pure polymer, this signifies its partially amorphous structure. After the incorporation of WO_3_–GO into PEDOT, the WO_3_–GO diffraction peaks were seen due to the PEDOT matrix.

The particle diameter of the as-prepared nanomaterial was determined by the Scherrer formula.
(4)$$D=\frac{k\cdot \lambda }{\beta cos\theta }$$
where *k* is the Scherrer constant, and was considered as 0.9 in this work. *β* is the line broadening value at half of the maximum intensity (FWHM), which is expressed as Δ2*θ* in radians. The mean crystallite size calculated using the Scherrer equation was found to be 215 nm for the PEDOT@WO_3_–GO nanomaterial, which agrees well with the mean diameter calculated from TEM measurements.

The XPS in Figure 2 shows the shift of C1s binding energy of the WO_3_–GO, PEDOT and PEDOT@WO_3_–GO samples. The C1s spectrum of WO_3_–GO shows a major peak at 284.59 eV which is attributed to the bonding energy of C–C related to component GO [27]. The second shoulder peak at 286.05 eV is attributed to C–OH (epoxy/hydroxy). Another oxygen containing group, O–C=O, was present in very small concentrations at 288.43 eV. Moreover, the typical carbon spectra (C1s) of pure PEDOT is exhibited in a peak at 284.57 eV and is assigned to the C=C chain of α-PEDOT [28]. The peak at 284.97 eV corresponds to the C=C chain of β-PEDOT [28], and the peak at 287.21 eV is assigned to C–O/C–S chains [29]. In addition, the ternary nanocomposite was observed to contain numerous oxygen-containing functional groups because of the presence of the two components, PEDOT and WO_3_–GO together. Therefore, the area under the curve at 286.48 eV and also at 286.48 eV are larger for PEDOT@WO_3_–GO than for WO_3_–GO. These results are consistent with those reported in the literature.

The narrow scan spectrum of the W4f core level for samples is shown in Figure 3. For pristine WO_3_, the deconvoluted peak observed at 35.48 eV could be related to the W4f_7/2_ level, whereas the peak at 37.62 eV could correspond to the W4f_5/2_ level [30]; whereas these peaks shift to more positive values for the WO_3_–GO material, to become the binding energies of 35.98 eV and 38.09 eV, respectively. Likewise, the W4f XPS spectrum of PEDOT@WO_3_–GO also displays two peaks located at 36.03 eV and 38.15 eV corresponding to the existence of the W^6+^ oxidation state.

Figure 4a illustrates the nitrogen adsorption–desorption isotherm of the materials. All samples show type IV isotherms with a typical H_3_ type hysteresis loop according to the Brunauer classification, indicating the existence of textural meso-/microporosity. This is mainly due to the presence of slit-shaped pores, which are formed by the accumulation of crystal nanoparticles [32]. Moreover, the specific surface area (S_BET_) and pore volume (V_pore_) of the WO_3_ are 23.57 m^2^·g^−1^ and 0.07 cm^3^·g^−1^, respectively. Likewise, calculations reveal that the S_BET_ and V_pore_ of the WO_3_–GO samples are 41.82 m^2^·g^−1^ and 0.08 cm^3^·g^−1^, respectively. The increase in the surface area can be explained by the GO nanosheets having the largest surface area, mainly contributed by the GO nanosheets’ high surface area in the WO_3_–GO architecture (S_BET_ of GO is 63.31 m^2^·g^−1^). In previous studies, PEDOT-specific surface area was varied, depending on the synthesis technique or even the treatment of polymer after preparation. Sequeira et al. [35] stated that the S_BET_ and V_pore_ were 27 m^2^·g^−1^ and 0.09 cm^3^·g^−1^, respectively. Cheng et al. [36] reported that the S_BET_ was 58.86 m^2^·g^−1^. The S_BET_ differed considerably when changing the material used in the nanocomposite’s preparation. Accordingly, it was found that the formation of the PEDOT matrix increases the S_BET_ of WO_3_–GO to 103.92 m^2^·g^−1^ and the corresponding V_pore_ to 0.11 cm^3^·g^−1^. Therefore, the polymer backbone in the PEDOT@WO_3_–GO structure would presumably provide additional space and volume for the diffusion of ions during the electrochemical charging and discharging processes.

Figure 4b illustrates the TGA curves for samples. PEDOT@WO_3_–GO demonstrated higher thermal stability than pure polymer. The largest weight loss occurs at temperatures from 480 °C to 610 °C for both PEDOT and PEDOT@WO_3_–GO, due to the destruction of the polymer backbone and carbon skeleton at the same time. In addition, the weight loss of PEDOT@WO_3_–GO and WO_3_–GO was stabilized at temperatures from 25 °C to 900 °C at about 83.53% and 93.78%, respectively; which indicates that the amount of polymer loaded on the WO_3_–GO material was about 10.25 wt%. Likewise, these results indicate that the amount of GO sheets formed on the WO_3_ surface was about 6.22 wt%.

The TEM images of WO_3_, WO_3_–GO and PEDOT@WO_3_–GO are given in Figure 5. It can be observed that WO_3_ has various oval-shaped nanoparticles with a particle size distribution between 90 nm and 160 nm. It is worth mentioning that, owing to the flexible and two-dimensional sheet-like nature of graphene and its derivatives, they can easily be used to wrap or encapsulate oval nanoparticles. GO has been applied for the encapsulation of WO_3_ nanoparticles. WO_3_–GO possesses a number of advantages when compared to bare WO_3_, including less nanoparticle aggregation as well as the enhancement of electrical, electrochemical, and optical properties [37]. Specifically, owing to the characteristically strong negative charge of GO the encapsulation of WO_3_ by GO results in the suppression of aggregation, with a particle size distribution between 170 nm and 240 nm. Interestingly, it can be seen in Figure 5c that the WO_3_–GO material is adorned with well-distributed PEDOT for the ternary nanocomposite, which is beneficial for SC devices. WO_3_–GO sheets form aggregates with an average size of about 150~300 nm. The PEDOT matrix covers WO_3_–GO sheets; the size of the PEDOT@WO_3_–GO is 100~400 nm, which agrees well with the mean diameter calculated from XRD measurements.

### 3.2. Electrochemical Studies

We used cyclic voltammetry (CV), galvanostatic charge discharge (GCD) and electrochemical impedance spectroscopy (EIS) to characterize the electrochemical performances of electrodes. All analyses were carried out in a three–electrode testing system with 3 M KOH as the electrolyte. Figure 6a presents the CV curves of WO_3_, WO_3_–GO, PEDOT and PEDOT@WO_3_–GO materials recorded at a scanning rate of 10 mV·s^−1^, with a potential window ranging from −0.1 V to +1.0  V. The PEDOT and PEDOT@WO_3_–GO electrodes featured obvious redox peaks. The PEDOT@WO_3_–GO electrode possessed the largest specific capacitance among these CV curves. Additionally, the PEDOT@WO_3_–GO electrode featured two pairs of redox peaks, the first within the ranges 0.2–0.4 V [(1)/(1′)] due to the existence of PEDOT and the second between 0.6–0.7 V [(2)/(2′)] related to the WO_3_–GO material. The CV measurements for PEDOT@WO_3_–GO revealed that the separation between anodic and cathodic peaks is equal to 160 mV for the first pair and 70 mV for the second pair [3,32]. Contrarily, the CV curve of the WO_3_–GO electrode had a rectangular shape with angular forms in which the contribution of electrical double-layer capacitor and pseudocapacitance may be distinguished. In spite of this, the pattern exhibits a small pair of peaks that appeared within the range 0.55–0.65 V [(3)/(3′)]. Moreover, the first oxidation/reduction peaks observed for PEDOT@WO_3_–GO were absent on the CV curves of WO_3_–GO, suggesting that effective interaction of the ions led to an electrical double-layer capacitor [32]. Thus, both the WO_3_–GO and PEDOT species in the PEDOT@WO_3_–GO structure contributed to the pseudocapacitance. Accordingly, an electrode prepared from the PEDOT@WO_3_–GO material would exhibit reversible redox reactions and rate capabilities.

The capacitive behaviors of the electrodes prepared were examined by GCD in a three–electrode arrangement at 0.1 A·g^−1^ with a potential range from −0.1 to 1.0 V. As shown in Figure 6b, all GCD curves have the form of a triangle, suggesting that the SC has perfect properties. Generally, the discharge curve comprises two steps; the first is the electrical double-layer capacitor with a potential ranging from 1.0 to 0.4 V and the second is the electrical double-layer capacitor and pseudocapacitance with a potential ranging from 0.4 to −0.1  V. Moreover, the non-linearity in the GCD plots demonstrated the pseudocapacitance behavior of all electrodes, which agreed with the results achieved from the CV plots [25]. The specific capacitance value of the PEDOT@WO_3_–GO electrode (478.3 F·g^−1^) is superior to those of the PEDOT (28.6 F·g^−1^), WO_3_ (57.4 F·g^−1^) and WO_3_–GO (145.3 F·g^−1^) electrode nanomaterials at a current density of 1.0 A·g^−1^. In addition, the reported specific capacitance value of the PEDOT, WO_3_, WO_3_–GO and PEDOT@WO_3_–GO by the CV graphs at the scan rate of 20 mV·s^−1^ is 32.5 F·g^−1^, 60.7 F·g^−1^, 147.1 F·g^−1^ and 503.2 F·g^−1^, respectively. The PEDOT@WO_3_–GO is higher than that of the other samples, which is due to the common contribution of PEDOT and WO_3_–GO to the electrochemical process. The synergistic effect of polymer matrix and WO_3_–GO enhances the electrochemical activity and thereby increases the specific capacitance. On the other hand, the direct addition of WO_3_–GO during the polymerization of EDOT can make PEDOT uniformly coat the outside of the WO_3_–GO nanoparticles, forming a “shell-core structure”, so that the two have a better coordination at the nanoscale, and in nanocomposites. Thus, the material exerts a better synergistic effect and improves the electrochemical performance of the composite material [38].

Figure 6c presents the variation in CV measurements of PEDOT@WO_3_–GO at different scan rates ranging from 10 mV·s^−1^ to 100 mV·s^−1^ across the potential window from −0.1 V to +1.0 V. This curve exhibited a well-defined redox form, inferring the characteristic pseudocapacitance behavior resulted from likely faradic redox processes [39]. Additionally, from the shape of the CV patterns, this electrode is well preserved, even at a scan rate equal to 100 mV·s^−1^, indicating that this material has the best rate capacity of all the synthesized materials. The specific capacitance value decreases as the scan rate rises, mainly due to ion exchange storage. Moreover, the shape of the CV patterns were similar; however, the peak current increased as the scan rate grew, indicating the excellent rate efficiency and reversibility of the charge–discharge process of the electrode. Moreover, CV measurements of WO_3_–GO at different scan rates were performed and are shown in Figure 6d. We can see that the size of the CV pattern increases as the scanning speeds increase. In addition, the CV curves are closer to rectangles and more symmetrical, as well as the oxidation and reduction peaks, indicating that a pseudocapacitance action is also taking place.

Figure 7a,b depicts the typical GCD plots of the WO_3_–GO and PEDOT@WO_3_–GO electrodes at various current densities and at a scan rate of 10 mV·s^−1^. The calculated specific capacitance of PEDOT@WO_3_–GO at 0.1 and 5.0 A·g^−1^ were 478.3 and 382.5 F·g^−1^, respectively, showing a good rate performance. The specific capacitance decreases with increasing current density, which may be due to an insufficient time for electrolyte ions to diffuse into the pores at the high current density. Additionally, a comparative analysis of WO_3_–GO was performed concerning current densities, which showed a regular increment in the specific capacitance with a decrease in current densities. The detailed specific capacitance performance of two electrodes is displayed in Figure 7c. These data fully prove that the fabricated PEDOT@WO_3_–GO electrode has incredibly potent adaptability to a large current charge/discharge, which was able to be applied in high-power charge/discharge occasions [35].

Figure 7d shows the investigations on cycling stability of PEDOT@WO_3_–GO and WO_3_–GO electrodes using charge/discharge measurements at 0.1 A·g^−1^ in the potential window of −0.1  V to +1.0 V over 5000 cycles. The cyclic performance of prepared electrodes are very important factors to be considered in SCs. It is observed from the figure that the PEDOT@WO_3_–GO electrode showed only a 95.2% reduction after 1000 cycles. Nevertheless, an extra charging/discharging process improved the stability of capacitance, with almost 92.1% of the capacitance remaining after the 5000-cycle test, confirming a very strong cycling stability. This demonstrates that the developed material is a reliable electrode for SC applications. The cyclic stability of WO_3_–GO showed as an 83.7% capacity retention at 5000 cycles. It is worth mentioning that the enhanced electrochemical performance of PEDOT@WO_3_–GO is due to the uniform distribution of the 2D layered PEDOT matrix and WO_3_ on 2D layered GO sheets. Hence, these multilayers on WO_3_–GO surfaces could be predicted to be a 3D layered structure, which could be caused by the introduction of PEDOT as a nitrogen source into the WO_3_–GO lattice, solving the restacking problem of GO sheets bonded to WO_3_ [5,40,41].

Figure 8a shows the curves of energy and power density for electrodes. It is observed that PEDOT@WO_3_–GO showed the highest value of the power density (971 W·kg^−1^) and energy density (54.2 Wh·kg^−1^) as compared to WO_3_–GO (power density is 585 W·kg^−1^ and power density is 29.7 Wh·kg^−1^). Thus, it confirms that PEDOT@WO_3_–GO showed enhanced charge transfer kinetics at the electrode/electrolyte interface with more stability towards electrochemical performance and hence, it could act as an alternative electrode material in SC applications. Table 1 compares the electrochemical performance of a few related electrodes; these results suggest the superiority of the PEDOT@WO_3_–GO electrode with those of other MO-based and GO-based materials.

Figure 8b shows the Nyquist plot of PEDO@WO_3_–GO and WO_3_–GO electrodes, the plot consists of a semicircle followed by a straight line in the low-frequency region, which is related to the electrochemical and mass transfer processes, respectively. In the low-frequency region, the quasi-vertical curves indicating the diffusion or Warburg impedance are mainly governed by both the contributions, i.e., non-faradaic and faradaic. From Figure 8b, it is observed that two electrodes make a transition angle between 65° and 70°, which is a clear indication of kinetics and diffusion of ions due to double-layer capacitive and faradaic behavior. In theory, the low-frequency region exhibiting an ideal capacitor should display vertical line behavior. The plot, however, demonstrates that the vertical line is slanted at an angle, which is associated with ion diffusion behavior. This deviation is caused by two factors; firstly, a variable penetration depth of an AC signal due to pore size dispersion at both electrode materials, resulting in anomalous capacity; and secondly, faradaic contribution occurring at the electrode surface [51]. In order to interpret the data collected from EIS, a Randles cell as an equivalent circuit was utilized and is shown in the inset Figure 8b, where (R_s_) is the resistance of the solution and (Z_W_) is the Warburg impedance for the diffusion of redox. (C_dl_) is the double-layer capacitance, and (R_ct_) is the charge-transfer resistance. For the PEDOT@WO_3_–GO electrode, the R_ct_ was about 9.7 Ω, which is less than that of WO_3_–GO material showing a R_ct_ of about 17.8 Ω. The Z_W_ of the PEDOT@WO_3_–GO, which appeared in the low frequency region and corresponded to a diffusion-controlled process, was much less than that of WO_3_–GO, indicating the abundance of ions on the electrode surface causing a decrease in impedance. In terms of the C_dl_, there was no difference between the two electrodes. Furthermore, the values of the equivalent series resistance (ESR), which include the electrolyte resistance, the internal electrode resistance, and the electrical resistance between the electrode and the current collector, can be obtained from the real axis intercept of the complex-plane impedance plots (inset of Figure 8b). At the same time, EDR is defined as the difference between R (the intersection value of the extrapolation of the EIS near-vertical linear segment and Z′-axis) and the ESR [52,53]. The PEDOT@WO_3_–GO electrode shows the lowest ESR of 0.34 Ω and EDR of 8.21 Ω in comparison with the WO_3_–GO material (ESR = 0.67 Ω, EDR = 17.82 Ω).

## 4. Conclusions

Formation of WO_3_–GO on a PEDOT matrix as a high-performance SC electrode material consisted of preparation of GO using a modified Hummer’s method and WO_3_ nanoparticle loading by an electrostatic self-assembly process to form a binary WO_3_–GO hybrid, has been accomplished. This was then followed by an in situ polymerization method of an EDOT monomer on the surface of WO_3_–GO. The preparation of samples was confirmed by XRD, FTIR, TGA, TEM, BET and XPS studies. The prepared PEDOT@WO_3_–GO electrode exhibited high cyclic stability of the material as described by the CV curve. EIS studies demonstrated high electrochemical performance with superior capacitive behavior. The highest specific capacitance (478.3 F·g^−1^), energy density (54.2 Wh·kg^−1^), power density (971 W·kg^−1^), and capacitance retention of 92.1% even after 5000 cycles was obtained for PEDOT@WO_3_–GO rather than the WO_3_–GO electrode (345.3 F·g^−1^, 29.7 Wh·kg^−1^ and 585 W·kg^−1^, respectively). These results showed that the multilayered ternary nanocomposite has high stability in terms of electrochemical performance and is considered a good choice of electrode for energy storage applications.

## Figures and Tables

**Figure 1 nanomaterials-13-02664-f001:**
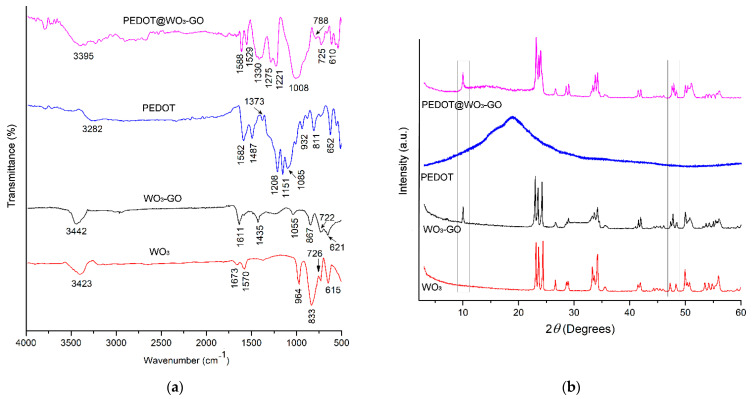
(**a**): FTIR analysis; and (**b**): X-ray diffraction (XRD) patterns of materials.

**Figure 2 nanomaterials-13-02664-f002:**
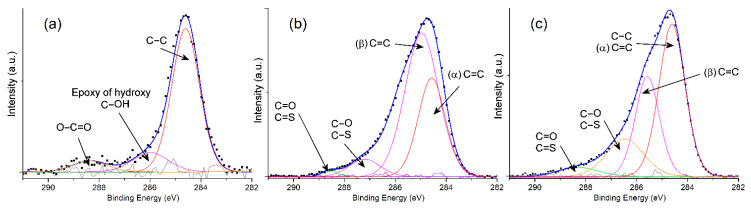
C1s scanning spectra of samples: (**a**) WO_3_–GO; (**b**) PEDOT and (**c**) PEDOT@WO_3_–GO.

**Figure 3 nanomaterials-13-02664-f003:**
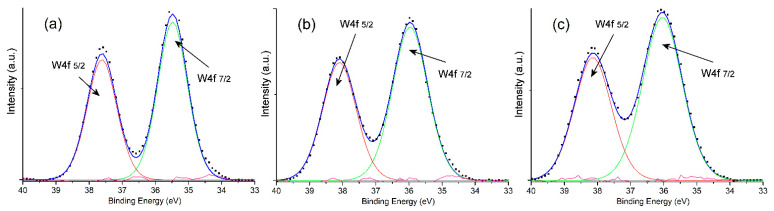
W4f scanning spectra of samples: (**a**) WO_3_; (**b**) WO_3_–GO and (**c**) PEDOT@WO_3_–GO.

**Figure 4 nanomaterials-13-02664-f004:**
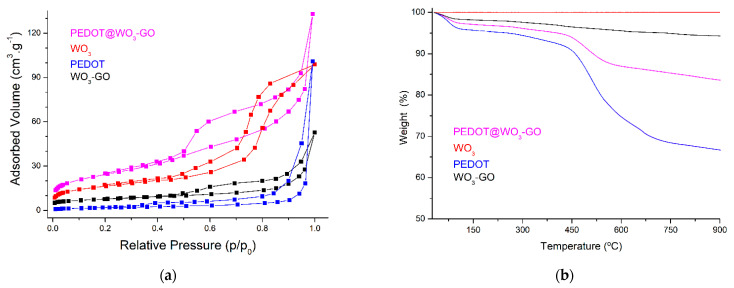
(**a**): Nitrogen adsorption isotherms; and (**b**): TGA curves of materials.

**Figure 5 nanomaterials-13-02664-f005:**
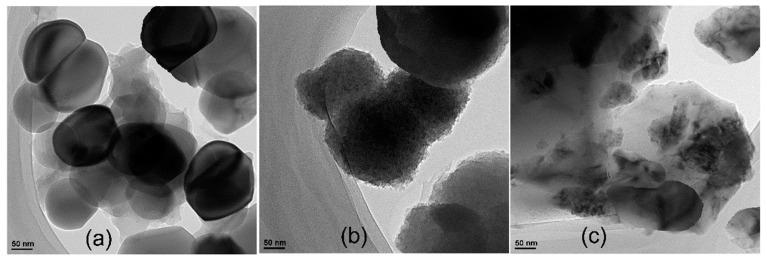
TEM images of: (**a**) WO_3_; (**b**) WO_3_–GO; and (**c**) PEDOT@WO_3_–GO.

**Figure 6 nanomaterials-13-02664-f006:**
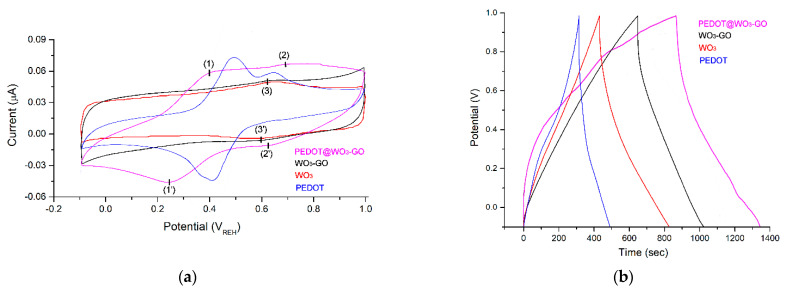
(**a**): CV curves of the electrodes prepared, recorded at 10 mV·s^−1^; (**b**): GCD of the electrodes prepared at current density of 0.1 A·g^−1^; (**c**): CVs of PEDOT@WO_3_–GO at different scan rates; and (**d**): CVs of WO_3_–GO at different scan rates.

**Figure 7 nanomaterials-13-02664-f007:**
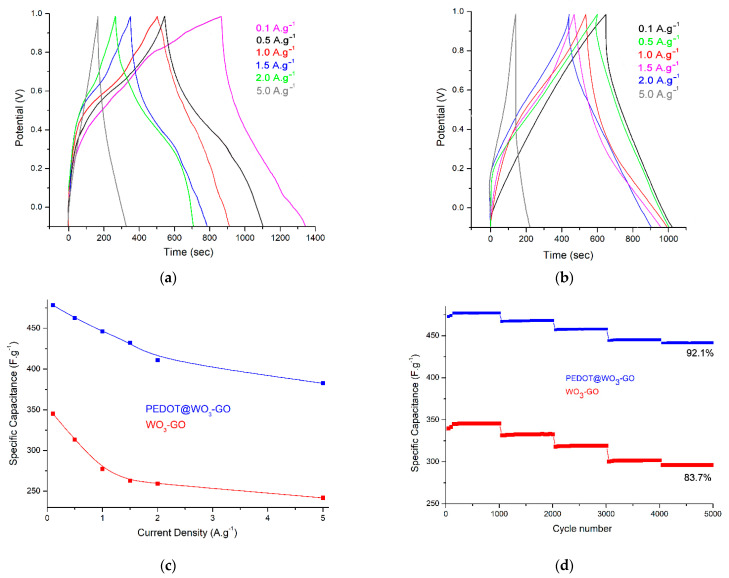
(**a**): GCD of PEDOT@WO_3_–GO; (**b**): GCD of WO_3_–GO at various current densities and at a scan rate of 10 mv·s^−1^; (**c**): Specific capacitance curves of electrodes at different current densities; and (**d**): Cycling stability of electrodes: specific capacitance versus cycle number measured at a current density of 0.1 A·g^−1^.

**Figure 8 nanomaterials-13-02664-f008:**
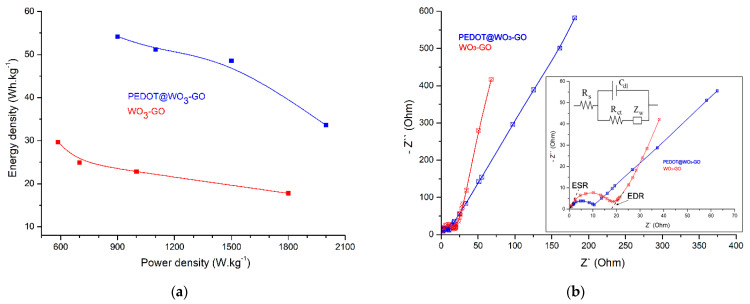
(**a**): Ragone plot obtained for electrode materials; and (**b**): Typical Nyquist plots of electrodes with the Randles equivalent circuit in the inset figure.

**Table 1 nanomaterials-13-02664-t001:** Electrochemical performance of different materials as supercapacitor electrodes.

Electrodes	Specific Capacitance (F·g^–1^)	Energy Density (Wh·kg^–1^)	Power Density (W·kg^–1^)	References
Graphene/Fe_2_O_3_	378.7	64.09	800.01	[13]
WO_3_/GO	143.6	//	//	[21]
WO_3_	32.4	//	//	[21]
Graphene–WO_3_ nanowire	465	25	6000	[21]
WO_3_/Se(ASC)	0.858	0.047	0.345	[42]
WO_3_–MnO_2_	103	24.13	915	[43]
WO_3_/SnO_2_	530	35	468	[44]
Graphene/PEDOT	270	34	25000	[45]
Graphene/SnO_2_/PEDOT	183	22.8	238.4	[46]
GO/PEDOT:PSS	155	10.79	1.53	[47]
PEDOT:PSS/MnO_2_/GO	841	593	//	[48]
GO/Glucose/PEDOT: PSS	19.72	//	//	[49]
Graphene/PEDOT:PSS/Ecoflex	82.4	11.44	131.58	[50]
WO_3_–GO	345.3	29.7	585	This work
PEDOT@WO_3_–GO	478.3	54.2	971	This work

## Data Availability

Data will be made available on reasonable request.

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
