# Peer review of "Fabrication and Characterization of a Poly(3,4-ethylenedioxythiophene)@Tungsten Trioxide–Graphene Oxide Hybrid Electrode Nanocomposite for Supercapacitor Applications"

_nanomaterials, 2023, doi:10.3390/nano13192664_

Round 1
Reviewer 1 Report
The manuscript "Fabrication and characterization of PEDOT@WO3-GO hybrid electrode material for supercapacitor application" reports the WO3-GO hybrid prepared using the electrostatic self-assembly process. This electrode shows more excellent performance in application compared with the previous studies, and this work is interesting and meaning for the readership of nanomaterials. However, the whole writing is very casual and the attitude is not correct, in which there are many obvious problems appear, if the following problems will be revised can be considered to receive the publication:
1) In the abstract section, the authors should briefly introduce the background of this work, and then mention the experimental scheme, important conclusions and significance. Please rewrite it.
2) The content of the introduction is somewhat confusing, and the literature review section suffers from summarizing without criticizing, so the authors need to reorganize and reform this section, and more importantly, to highlight the innovation of this work.
3) Punctuations should be consistent between abbreviations and brands in the material in the experimental section, such as “Potassium permanganate (KMnO4; Merck), Ethanol (C2H5OH) (Merck, 96%)”. There should be a space between the number and the unit, such as 20ËšC, 5g. And the format of all figure legend should be uniform. These are some non-negligible issues that require careful checking and revision by the authors.
4) The legend of Figure 3is wrong. And are the results present in Figure 3b correct (can W be measured in PEDOT)?
5) The description “the WO3-GO” material exhibited a notable shell-core structure and their core and shell were WO3 and GO, respectively.” The authors should provide other evidence of what the composition of the core and shell are, respectively.
6) The color and symbol of the figure when the same samples or conditions should be consistent so that the figure becomes more understandable.
7) In addition to the commonly used electrochemical analyses such as CV, GCD, electrochemical impedance spectroscopy (EIS) is also necessary. The authors should add the results of EIS to analyze the excellent electrochemical performance exhibited by the prepared materials. Please refer to the classical papers on Nyquist complex impedance spectrum of electrodes such as Carbon 185 (2021) 630e640.
the whole writing is very casual.
Author Response
Response to Reviewer 1 Comments
Point 1: In the abstract section, the authors should briefly introduce the background of this work, and then mention the experimental scheme, important conclusions and significance. Please rewrite it.
Response 1: We reformulated the Abstract (see lines 19-25).
“With the rapid development of nanotechnology, the study of nanocomposites as electrode materials has significantly enhanced the scope of research towards energy storage applications. Exploring electrode material with superior electrochemical properties is still a challenge for high-performance supercapacitors. In the present research article, we are prepared a novel nanocomposite of WO3 nanoparticles grown over supported Graphene oxide (GO) sheets and embedded with PEDOT matrix to maximize its electric double layer capacitance. The structural and morphological analysis of prepared electrodes were extensively characterized by XRD, FTIR, TEM, TGA, XPS, TGA and BET.”
Point 2: The content of the introduction is somewhat confusing, and the literature review section suffers from summarizing without criticizing, so the authors need to reorganize and reform this section, and more importantly, to highlight the innovation of this work.
Response 2: We reformulated the introduction (see lines 46-51, lines 53-58, lines 81-83 and lines 105-126)
“Moreover, some of the electrode materials studied for effective storage of charge includes the transition metals, carbonaceous material, and conductive polymeric materials. For use in SC applications, carbonaceous electrode materials like mesoporous carbon, graphene oxide (GO), reduced graphene oxide, carbon nanofibre, activated carbon, and carbon nanotubes exhibit outstanding rate capability, good reversibility, and excellent stability.”
And
“Furthermore, GO is a flexible lamellar material that has a wide range of functional groups on both basal planes and edges. As a result, it can be easily exfoliated and functionalized to form homogeneous suspensions in both water and organic solvents, providing more possibilities for the synthesis of graphene-based materials. The existence of oxygen functional groups and aromatic sp2 domains allows GO to participate in a wide range of bonding interactions.”.
And
“Because of its 2D layered structure as like GO sheets, which makes high reversibility of the intercalation of ions, high electronic conductivity and large current capacity [5]”
And
“As recently reported, PEDOT, GO and nanocomposites of WO3 and conducting polymers can effectively improve the power density of supercapacitors [26-32]. However, the aggregation of nanomaterials easily occurs during the construction of 3D nanostructures when using the abovementioned materials, leading to a decrease in the specific surface area [33]. In the literature, studies have been found on the contribution of CP to GO with presence of metal oxide [1-3, 11-15]. Haldar et al. [1] produced via chemical polymerization method the metal oxide/PANI/Graphene composites using different metal oxides (ZrO2, WO3 and V2O5) and investigated its electrochemical properties. Bejjanki et al. [34] synthetized SnO2/RGO@PANi ternary composite via chemical oxidation polymerization for supercapacitors. According to their results, they observed that adding metal oxide to the GO with presence of CP matrix greatly affects the supercapacitor properties and increases the specific capacitance value. These results inspired our study.
In this work, we report a simple method to prepare PEDOT@WO3−GO ternary composite starting with WO3−GO, which was considered as primary material where WO3 was in-situ on GO sheets using a simple chemical synthesis method. Secondly, polymer matrix are followed via chemical oxidation polymerization of EDOT. The PEDOT@WO3−GO electrode material so obtained has excellent electrochemical performance and good specific capacitance of 478.3 F·g–1 and a maximum energy density of 54.2 Wh·kg–1 with power density 971 W·kg–1, this is attributed to larger effective surface area of the fabri-cated materials These findings indicate that the PEDOT@WO3−GO nanocomposites could be promising electrode materials for high-performance supercapacitor applications, which was not previously reported.”
Point 3: Punctuations should be consistent between abbreviations and brands in the material in the experimental section, such as “Potassium permanganate (KMnO4; Merck), Ethanol (C2H5OH) (Merck, 96%)”. There should be a space between the number and the unit, such as 20ËšC, 5g. And the format of all figure legend should be uniform. These are some non-negligible issues that require careful checking and revision by the authors.
Response 3: We have reviewed and corrected all these comments.
Point 4: The legend of Figure 3is wrong. And are the results present in Figure 3b correct (can W be measured in PEDOT)?.
Response 4: We corrected the typo in Figure 3 (see line 265).
Point 5: The description “the WO3-GO” material exhibited a notable shell-core structure and their core and shell were WO3 and GO, respectively.” The authors should provide other evidence of what the composition of the core and shell are, respectively.
Response 5: The Reviewer 3 asked us to rephrase this paragraph, so it became as follows: (see lines 296-303).
“It is worth mentioning that, owing to the flexible and two-dimensional sheet-like nature of graphene and its derivatives, they can easily be used to wrap or encapsulate oval nanoparticles. GO has been applied for the encapsulation of WO3 nanoparticles. WO3−GO possess a number of advantages when compared to bare WO3 including less nanoparticle aggregation as well as the enhancement of electrical, electrochemical, and optical properties [37]. Specifically, owing to the characteristically strong negative charge of GO the encapsulation of WO3 by GO results in the suppression of aggregation, with a particle size distribution between 170 nm and 240 nm.”
Point 6: The color and symbol of the figure when the same samples or conditions should be consistent so that the figure becomes more understandable.
Response 6: We unified the colors and symbols in all Figures.
Point 7: In addition to the commonly used electrochemical analyses such as CV, GCD, electrochemical impedance spectroscopy (EIS) is also necessary. The authors should add the results of EIS to analyze the excellent electrochemical performance exhibited by the prepared materials. Please refer to the classical papers on Nyquist complex impedance spectrum of electrodes such as Carbon 185 (2021) 630e640.
Response 7: We added EIS results to analyze the excellent electrochemical performance depending on the reference (Carbon 185 (2021) 630-640. https://doi.org/10.1016/j.carbon.2021.09.059) (see lines 440-447).
“Furthermore, the values of the equivalent series resistance (ESR), which include the electrolyte resistance, the internal electrode resistance, and the electrical resistance between the electrode and the current collector, can be obtained from the real axis intercept of the complex-plane impedance plots (inset of Figure 8-b). At the same time, EDR is defined as the difference between R (the intersection value of the extrapolation of EIS near-vertical linear segment and Z′-axis) and the ESR [52, 53]. The PEDOT@WO3–GO electrode shows the lowest ESR of 0.34 Ω and EDR of 8.21 Ω in comparison with the WO3–GO material (ESR = 0.67 Ω, EDR = 17.82 Ω).”

Reviewer 2 Report

The English language and writing need major improvement.|
Author Response
Response to Reviewer 2 Comments
Point 1: Regarding structure characterization of PEDOT@WO3−GO, the authors only briefly analyzed the FTIR, XRD, XPS, TGA, and BET, the advantages of nanostructures are lacking. In fact, this work cannot satisfy the requirements of the “Nanomaterials” journal.
Response 1: The word (material) has been changed to the word (nanocomposite) in the title of this manuscript, and also we have calculated the average size of PEDOT@WO3−GO to prove that the synthetized samples are nanomaterials type (see the title, lines 232-239 and lines 305-308):
“The particle diameter of the as-prepared nanomaterial was determined by the Scherrer formula.
The k is Scherrer constant, was considered as 0.9 in this work. ? is the line broadening value at half of the maximum intensity (FWHM), which is expressed as Δ2? in radians. The mean crystallite size calculated using the Scherrer equation was found to be 215 nm for PEDOT@WO3−GO nanomaterial, which agrees well with the mean diameter calculated from TEM measurements.”
And
“WO3−GO sheets form aggregates with average size of about 150∼300 nm. The PEDOT matrix covers WO3−GO sheets, the size of the PEDOT@WO3−GO is 100∼400 nm, which agrees well with the mean diameter calculated from XRD measurements.”
Point 2: The discussions about the physicochemical properties and the electrochemical performances of PEDOT@WO3−GO in the manuscript are not sufficient. The authors only briefly presented the results, lacking a deep discussion. For example, in the 3.2 Electrochemical studies section, the authors only claimed the PEDOT@WO3−GO electrode shows the highest specific capacitance (478.3 F.g−1) than PEDOT (28.6 F.g−1), WO3 (57.4 F.g−1), and WO3−GO (145.3 F.g−1) electrode materials. The authors should give a detailed explanation.
Response 2: We have added a detailed discussion about these results. (See lines 346-353)
“The PEDOT@WO3−GO is higher than that of other samples, which is due to the common contribution of PEDOT and WO3−GO to the electrochemical process. The synergistic effect of polymer matrix and WO3−GO enhances the electrochemical activity and thereby increases the specific capacitance. On the other hand, the direct addition of WO3−GO during the polymerization of EDOT can make PEDOT uniformly coat the outside of the WO3−GO nanoparticles, forming a “shell-core structure”, so that the two have a better coordination on the nanoscale, and in nanocomposite. Thus, the material exerts a better synergistic effect and improves the electrochemical performance of the composite material [33].”
Point 3: As the authors claimed, “the enhanced electrochemical performance of PEDOT@WO3−GO is due to the uniform distribution of two-dimensional (2D) layered PEDOT matrix and WO3 on 2D layered GO sheet. Hence, these multi-layers on WO3−GO surface could be predicted as three-dimensional (3D) layered structure.” Please provide the evidences.
Response 3: We added an explanation that supports the proposed possibility of the formation of the 2D and 3D structures for WO3−GO and PEDOT@WO3−GO samples, respectively (see lines 81-83 and lines 403-404).
“Because of its 2D layered structure as like GO sheets, which makes high reversibility of the intercalation of ions, high electronic conductivity and large current capacity [5].”
And
“which can be caused by the introduction of PEDOT as a nitrogen source into WO3−GO lattice , solving the restacking problem of GO sheets bonded to WO3 [5, 40, 41].”
Point 4: The authors only introduce one PEDOT@WO3−GO composite sample, lacking comparison of other controlled samples with different amount of GO, WO3, and EDOT. A sample did not reveal scientific nature. The authors should clearly explain the novelty of the ternary nanocomposite in the manuscript.
Response 4: We have added a sentence that clearly explains the novelty of ternary nanocomposite in supercapacitor applications (see lines 105-109). Moreover, various research has been studied before regarding (GO, WO3, and PEDOT) materials as capacitors, as described in references [20], [21], [24], [32], [43]. Additionally, we tested several experiments with changing the percentage of the PEDOT in the WO3-GO, and all the results were very similar. On the other hand, we are in the process of preparing a new work based on the use of different amounts of PANI with same WO3-GO amount.
“As recently reported, PEDOT, GO and nanocomposites of WO3 and conducting polymers can effectively improve the power density of supercapacitors [26-32]. However, the aggregation of nanomaterials easily occurs during the construction of 3D nanostructures when using the abovementioned materials, leading to a decrease in the specific surface area [33].”
Point 5: Supercapacitors have been widely used as high-power devices due to their fast charge-discharge ability and long lifetime. The authors only provide specific capacitance at a current density of 0.1~2 A.g−1 in Figure 7 (c). In general, metal oxides electrode materials can achieve excellent capacitances at the current density more than 10 A.g−1. Please provide specific capacitances of PEDOT@WO3−GO and WO3−GO at larger current densities.
Response 5: Yes, you are right, we have added larger current capacity results up to 5.0 A.g−1 (See Figure 7-a, 7-b and 7-c). However, we apply a protocol in our laboratory, and this protocol is consistent with several researches published in this journal and others, for example:
- Enhancement of Electrochemical Performance of Aqueous Zinc Ion Batteries by Structural and Interfacial Design of MnO2 Cathodes: The Metal Ion Doping and Introduction of Conducting Polymers. Energies 2023, 16(7), 3221; https://doi.org/10.3390/en16073221.
- Vanadium Oxide–Conducting Polymers Composite Cathodes for Aqueous Zinc-Ion Batteries: Interfacial Design and Enhancement of Electrochemical Performance. Energies 2022, 15(23), 8966; https://doi.org/10.3390/en15238966.
- Polyaniline-Coated Porous Vanadium Nitride Microrods for Enhanced Performance of a Lithium–Sulfur Battery. Molecules 2023, 28(4), 1823; https://doi.org/10.3390/molecules28041823.
- Effects of Carbon Content and Current Density on the Li+ Storage Performance for MnO@C Nanocomposite Derived from Mn-Based Complexes. Nanomaterials 2020, 10, 1629. https://doi.org/10.3390/nano10091629.
- Carbon Nanocluster-Mediated Nanoblending Assembly for Binder-Free Energy Storage Electrodes with High Capacities and Enhanced Charge Transfer Kinetics. 2023, 10, 2301248. https://doi.org/10.1002/advs.202301248.
- Improving the Reaction Kinetics by Annealing MoS2/PVP Nanoflowers for Sodium-Ion Storage. Molecules 2023, 28, 2948. https://doi.org/10.3390/molecules28072948.
Point 6: In Figure 7 (a) and (b), the charge and discharge time in GCD are different, which will result in strong polarization and the low coulombic efficiency of materials lower than 100%. Besides, low coulombic efficiency cannot meet the good rate reversibility for capacitive energy storage.
Response 6: Yes, this problem can is the presece of adhesives and conducting fillers as CB and PVDF, which are necessary for these electrode materials. its presence not only results in a complicated preparation process, but also in an increase in contact resistance and a decrease in the capacitance performance, according to what was ivestigated by Huang et. al. [33].
Point 7: As the authors claimed, “Figure 6 (a) depicts the CV curves of WO3, WO3−GO, PEDOT and PEDOT@WO3−GO materials recorded at a scanning rate of 5mV.s−1 with potential range from −0.1V to +1.0 V.” The scan rate is not consistent with the notes of Figure 6 (a).
Response 7: Yes, it was a typo and we corrected it; the scan rate value is 10 mV.s−1 (See line 281 and line 317)
Point 8: In 2.5 Synthesis of PDOT@ZrO2−GO, the title and content are not consistent.
Response 8: Yes, it's a typo and we corrected it as follows (See line 168 and line 317):
2.5. Fabrication of PEDOT@WO3−GO
Point 9: Figure 1, Figure 2, and Figure 3 lack ordinates. The authors should beautify the Figures in the manuscript.
Response 9: We added the coordinates on the axes of Figure 1, Figure 2, and Figure 3.
Point 10: The expressions and formats of the manuscript need to be checked carefully. For example, the caption in Figure 7 (a) and (b) are confusing; “PEDO@WO3−GO” in Table 1 is missing a letter “T”.
Response 10: Yes, it was a typo and we corrected itTable 1)
Point 11: As the author presented “capacitance of 478.3F·g−1 at 50mV.s−1 as comparedto WO3−GO (345.3F·g−1 )”, “The retention capacity of92.1% up to 5000 cycles at 0.1 A.g−1 ”, “hysteresis loopaccording to Brunauer classification”, “GCD of WO3-GO at different current densities and at scan rate of 50mv.s-1 ” etc. Missing space between number and unit.
Response 11: We corrected all obsevations related to missing space between number and unit.
Point 12: There are some format errors in the cited references such as the abbreviations of the journal name are not consistent, writing format is not uniform etc. Please unify reference formats.
Response 12: The formatting of the references will be done by the journal team as mentioned in the "Instructions for Authors" for this journal. However, we have unifyed the reference formats in manuscript.

Reviewer 3 Report
Dear Sir,
The authors reported “Fabrication and characterization of PEDOT@WO3−GO hybrid electrode material for supercapacitor application”. But authors did not give the brief and meaningful explanation in abstract; therefore, it is needed to modify. I, therefore, feel that this manuscript can be published in the esteemed journal ‘Nanomaterials (ISSN 2079-4991)’ after the Major Revision.
Comments-
1. The abstract should be eye-catching, incorporate the novelty and finding of the work briefly.
2. Please check the writing very carefully, in some sentences the main subject is not clear.
3. Some section in introduction is not clear please check carefully, such as 1. ‘Moreover, some of the electrodes stud- 40 ied for succeeded storage of charge includes hydroxides, nitrides, sulphides and oxides 41 of conductive polymeric, transition metals and carbon materials. For employ in SC appli- 42 cations, carbonaceous electrodes like activated carbon, carbon nanotubes, mesoporous 43 carbon, activated carbon, carbon nanofibre, reduced graphene oxide (rGO) and graphene 44 oxide (GO) present excellent rate capability’ In this following sentence many grammatical errors. Please check carefully. In line, 47 to 49, the dscription of GO is not sufficient and wrongly interpreted, please check again.
In GO all carbon is not Sp2 hybridized.
4. Authors should mention the importance of synthetic method and cite the recent articles.
5. Authors should mention the details of synthetic procedure e.g. molar concentration of individual precursor, provide some good quality of photos of the synthesized materials for clear idea.
6. In line 137, author mention ZrO2, but the material is WO3, please make sure which material author used to synthesize
7. In XRD analysis, the author should mention the formulae and proper citation of the crystallite size calculation
8. The author should check the image quality, it does not have enough clarity.
9. In Table one some of the words are not in English, please check
10. The bar line of TEM images is not clear, please provide the clear bar line of individual images, images are also not clear.
11. Please check all images, almost difficult to understand
12. Please provide the details surface area analysis
13. In electrochemical analysis the mechanism is not clear please rewrite thoroughly
14. Please provide the details of electrochemical formulae that are used to calculate the capacitance, power, and energy density.
15. Please check the line 253 to 257, the author did not mention why PEDOT is faradaic in nature. Please provide some equations, so the reader can understand properly what is going on in the electrochemical system.
16. Please provide the circuit diagram ion EIS spectra, write down the details analysis
17. Please check the overall manuscript with an English expert. In the manuscript, the subject is not clear what the author would like to say
18. Please check the comparison table, no need to put WO3-rGO results, it's not your final materials.
please check it very carefully, most of the lines are rearranged, which misleads the scientific soundness. Thanks
Author Response
Response to Reviewer 3 Comments
Point 1: The abstract should be eye-catching, incorporate the novelty and finding of the work briefly.
Response 1: We reformulated the Abstract (see lines 19-25).
“With the rapid development of nanotechnology, the study of nanocomposites as electrode materials has significantly enhanced the scope of research towards energy storage applications. Exploring electrode material with superior electrochemical properties is still a challenge for high-performance supercapacitors. In the present research article, we are prepared a novel nanocomposite of WO3 nanoparticles grown over supported Graphene oxide (GO) sheets and embedded with PEDOT matrix to maximize its electric double layer capacitance. The structural and morphological analysis of prepared electrodes were extensively characterized by XRD, FTIR, TEM, TGA, XPS, TGA and BET.”
Point 2: Please check the writing very carefully, in some sentences the main subject is not clear.
Response 2: We checked the writing very carefully, so that the sentences are clear.
Point 3: Some section in introduction is not clear please check carefully, such as 1. ‘Moreover, some of the electrodes studied for succeeded storage of charge includes hydroxides, nitrides, sulphides and oxides of conductive polymeric, transition metals and carbon materials. For employ in SC applications, carbonaceous electrodes like activated carbon, carbon nanotubes, mesoporous carbon, activated carbon, carbon nanofibre, reduced graphene oxide (rGO) and graphene oxide (GO) present excellent rate capability’ In this following sentence many grammatical errors. Please check carefully. In line, 47 to 49, the dscription of GO is not sufficient and wrongly interpreted, please check again.
In GO all carbon is not Sp2 hybridized.
Response 3: We reformulated the sentences (see lines 46-51, lines 53-58).
“Moreover, some of the electrode materials studied for effective storage of charge includes the transition metals, carbonaceous material, and conductive polymeric materials. For use in SC applications, carbonaceous electrode materials like mesoporous carbon, graphene oxide (GO), reduced graphene oxide, carbon nanofibre, activated carbon, and carbon nanotubes exhibit outstanding rate capability, good reversibility, and excellent stability.”
And
“Furthermore, GO is a flexible lamellar material that has a wide range of functional groups on both basal planes and edges. As a result, it can be easily exfoliated and functionalized to form homogeneous suspensions in both water and organic solvents, providing more possibilities for the synthesis of graphene-based materials. The existence of oxygen functional groups and aromatic sp2 domains allows GO to participate in a wide range of bonding interactions.”
Point 4: Authors should mention the importance of synthetic method and cite the recent articles.
Response 4: We added this paragraph in order to point out the importance of the synthetic method used (see lines 109-116).
“In the literature, studies have been found on the contribution of CP to GO with presence of metal oxide [1-3, 11-15]. Haldar et al. [1] produced via chemical polymerization method the metal oxide/PANI/Graphene composites using different metal oxides (ZrO2, WO3 and V2O5) and investigated its electrochemical properties. Bejjanki et al. [34] synthetized SnO2/RGO@PANi ternary composite via chemical oxidation polymerization for supercapacitors. According to their results, they observed that adding metal oxide to the GO with presence of CP matrix greatly affects the supercapacitor properties and increases the specific capacitance value. These results inspired our study.”
Point 5: Authors should mention the details of synthetic procedure e.g. molar concentration of individual precursor, provide some good quality of photos of the synthesized materials for clear idea.
Response 5: We added information to clarify the details of synthetic procedure as described in (2.5. Fabrication of PEDOT@WO3−GO) section
.
Point 6: In line 137, author mention ZrO2, but the material is WO3, please make sure which material author used to synthesize.
Response 6: Yes, it's a typo and we corrected it as follows (See line 159):
2.5. Fabrication of PEDOT@WO3−GO.
Point 7: In XRD analysis, the author should mention the formulae and proper citation of the crystallite size calculation.
Response 7: We added this paragraph to clarify the method for calculating the crystallite size (see lines 223-230).
The particle diameter of the as-prepared nanomaterial was determined by the Scherrer formula.
The k is Scherrer constant, was considered as 0.9 in this work. ? is the line broadening value at half of the maximum intensity (FWHM), which is expressed as Δ2? in radians. The mean crystallite size calculated using the Scherrer equation was found to be 215 nm for PEDOT@WO3−GO nanomaterial, which agrees well with the mean diameter calculated from TEM measurements.
Point 8: The author should check the image quality, it does not have enough clarity.
Response 8: We added the coordinates on axes and enlarged the Figure so that the image quality is clear,
Point 9: In Table one some of the words are not in English, please check.
Response 9: We corrected a typo in the table 1.
Point 10: The bar line of TEM images is not clear, please provide the clear bar line of individual images, images are also not clear.
Response 10: We enlarged Figure 5 so that the image quality and bar line are clear.
Point 11: Please check all images, almost difficult to understand.
Response 11: We increased quality images are at scale (3500x3400).
Point 12: Please provide the details surface area analysis.
Response 12: We provided more details on surface area analysis (see lines 142-146).
“Surface area and pore volume analyses were performed on an iQinstrument Autosorb analyzer (Madrid, Spain). Prior to measurements, the samples were degassed at 120 °C under high vacuum overnight. The Brunauer–Emmett–Teller (BET) method was used to calculate the specific surface of the materials. The pore size distribution was derived from Barret–Joyner–Halenda (BJH).”
Point 13: In electrochemical analysis the mechanism is not clear please rewrite thoroughly.
Response 13: We rewritten the electrochemical analysis section thoroughly (see lines 184-187)
”To prepare WE, active material of 70 wt%, CB of 15 wt% and PVDF of 15 wt%, were mixed in acetone and stirred at 60 °C until a homogeneous ink was obtained. Subsequently, the ink was drop-casted on stainless-steel (SS) plate and dried at 60 °C overnight.”
Point 14: Please provide the details of electrochemical formulae that are used to calculate the capacitance, power, and energy density.
Response 14: We provided details of the electrochemical formulas used to calculate the capacitance, power, and energy density (see lines 188-194).
“The specific capacitance Csp (F.g−1) from the electrode setup was calculated from CV and GCD by Equations 1 and 2:
|
(1) |
|
|
(2) |
The power density (E) and energy density (P) were calculated using galvanostatic discharge behavior via Equations (3) and (4):
|
(3) |
|
|
(4) |
Where Csp denotes specific capacitance, refers to the potential window, v is the scan rate, m is mass of the active material, I (A) is applied current and (s) is discharge time.”
Point 15: Please check the line 253 to 257, the author did not mention why PEDOT is faradaic in nature. Please provide some equations, so the reader can understand properly what is going on in the electrochemical system.
Response 15: We have rephrased the sentence so it is clear (see lines 322-324 and lines 327-329):
“The CV measurements for PEDOT@WO3−GO revealed that the separation between anodic and cathodic peaks is equal to 160 mV for the first pair and 70 mV for the second pair”
And
“Moreover, the first oxidation/reduction peaks observed for PEDOT@WO3−GO were absent on the CV curves of WO3−GO”
Point 16: Please provide the circuit diagram ion EIS spectra, write down the details analysis.
Response 16: We have provided the circuit diagram ion EIS spectra as shown in inset Figure 8-b, and we added the following phrase (see lines 431-440):
“In order to interpret the data collected from EIS, Randles cell as an equivalent circuit proposed was utilized, shown in the inset Figure 8-b, where (Rs) is the resistance of the solution, (ZW) is the Warburg impedance for the diffusion of redox. (Cdl) is the double-layer capacitance, and (Rct) is the charge-transfer resistance. For the PEDOT@WO3−GO electrode, the Rct was about 9.7 Ω, which is less than that of WO3−GO material showing the Rct about 17.8 Ω. The ZW of the PEDOT@WO3−GO, which appeared in the low frequency region and corresponded to diffusion-controlled process, was highly less than that of WO3−GO, indicating the abundance of ions on electrode surface causing a decrease in impedance. In term of the Cdl, there was no difference between the two electrodes.”
Point 17: Please check the overall manuscript with an English expert. In the manuscript, the subject is not clear what the author would like to say.
Response 17: We rechecked the manuscript in English.
Point 18: Please check the comparison table, no need to put WO3-rGO results, it's not your final materials.
Response 18: We omitted the results for WO3-rGO from the comparison table (see Table 1)

Reviewer 4 Report
The manuscript titled "Fabrication and characterization of PEDOT@WO3−GO hybrid electrode material for supercapacitor application" compares various electrode material for supercapacitor applications. However, there is a lack of novelty, and some experimental data needs proper explanation.
Comments:
1) Many spelling and formatting typos in this paper and the authors should check and revise them thoroughly.
2) The authors should clearly state the novelty of their work. Since WO3/GO and the mentioned conducting polymer (CP) are already reported in references 21 to 24, 42 to 45, why are the authors repeating similar strategies?
3) The caption of Figure 3 (line 215) does not match that of Figure 3.
4) The authors discussed the electrochemical performance of PEDOT@WO3−GO hybrid electrode in 3M KOH at length (line 149-150), yet their experiments are in 1M H2SO4. The connection between the 1M H2SO4 study and the present 3M KOH study should be clarified since not much is apparent if one compares the electrochemical performance of PEDOT@WO3−GO hybrid electrode, other than that ion can influence the performance.
5) What are the thicknesses and structures of the PEDOT@WO3−GO hybrid electrode grown on the stainless-steel plate? The SEM images of the cross sections of the electrodes can help the readers to understand the structure of the hybrid electrode.
6) The authors should analyze and discuss the role of porous structure in ion transport etc. The BET data should be carefully analyzed including the SSA, pore size distribution and their effects on the performance.
7) Why the author only presents the capacitive behaviors of the PEDOT@WO3−GO hybrid electrodes in a 3−electrode system? The performance of the ASC devices or SSC device (2−electrode system) is missing? Asymmetric supercapacitors (ASCs) assembled using two dissimilar electrode materials offer a distinct advantage of wide operational voltage window, and thereby significantly enhance the energy density.
8) Fit the EIS plots in Figure 8b with a proper equivalent circuit diagram and discuss the corresponding solution resistance, charge transfer resistance, and Warburg impedance.
Many spelling and formatting typos in this paper and the authors should check and revise them thoroughly.
Author Response
Response to Reviewer 4 Comments
Point 1: Many spelling and formatting typos in this paper and the authors should check and revise them thoroughly.
Response 1: We rephrased and corrected all formatting typos.
Point 2: The authors should clearly state the novelty of their work. Since WO3/GO and the mentioned conducting polymer (CP) are already reported in references 21 to 24, 42 to 45, why are the authors repeating similar strategies?.
Response 2: We reformulated the objectives of the research (see lines 117-126). Moreover, our method of preparing the material differs from the references (21 to 24 and 42 to 45), In addition, we used materials different from what was mentioned in those references and the values we obtained are higher.
“In this work, we report a simple method to prepare PEDOT@WO3−GO ternary composite starting with WO3−GO, which was considered as primary material where WO3 was in-situ on GO sheets using a simple chemical synthesis method. Secondly, polymer matrix are followed via chemical oxidation polymerization of EDOT. The PEDOT@WO3−GO electrode material so obtained has excellent electrochemical performance and good specific capacitance of 478.3 F·g–1 and a maximum energy density of 54.2 Wh·kg–1 with power density 971 W·kg–1, this is attributed to larger effective surface area of the fabri-cated materials These findings indicate that the PEDOT@WO3−GO nanocomposites could be promising electrode materials for high-performance supercapacitor applications, which was not previously reported.”
Point 3: The caption of Figure 3 (line 215) does not match that of Figure 3.
.Response 3: We corrected the typo in Figure 3 (see line 265).
Point 4: The authors discussed the electrochemical performance of PEDOT@WO3−GO hybrid electrode in 3M KOH at length (line 149-150), yet their experiments are in 1M H2SO4. The connection between the 1M H2SO4 study and the present 3M KOH study should be clarified since not much is apparent if one compares the electrochemical performance of PEDOT@WO3−GO hybrid electrode, other than that ion can influence the performance.
Response 4: Yes, you are right, we corrected the typo, the electrolyte is 3M KOH (see lines 301).
Point 5: What are the thicknesses and structures of the PEDOT@WO3−GO hybrid electrode grown on the stainless-steel plate? The SEM images of the cross sections of the electrodes can help the readers to understand the structure of the hybrid electrode.
Response 5: We added the thickness of the stainless steel plate (SS) used, and thicknesses of the PEDOT@WO3−GO hybrid electrode grown on the SS (see lines 142-146 and lines 187).
“A commercial grade stainless steel (SS) foil (thickness: 0.2 mm) was used as the substrate for electrodes preparation.”
Yes, you are right, but the problem is that the our SEM instrument is broken and we are on vacation, so the time does not allow for that.
Point 6: The authors should analyze and discuss the role of porous structure in ion transport etc. The BET data should be carefully analyzed including the SSA, pore size distribution and their effects on the performance.
Response 6: We reformulated and analyzed the BET data (see lines 271-284).
“Likewise, calculations reveals that the SBET and Vpore of the WO3−GO samples are 41.82 m2.g−1 and 0.08 cm3.g−1, respectively. The increase of the surface area can be explained the GO nanosheets had the largest surface area, mainly the GO nanosheets (SBET of GO is 63.31 m2.g−1) contributed the high surface area of the WO3−GO architecture. In previous studies, PEDOT-specific surface area was varied, depending on the synthesis technique or even the treatment of polymer after preparation. Sequeira et al. [35] stated that the SBET and Vpore are 27 m2.g−1 and 0.09 cm3.g−1, respectively. Cheng et al. [36] reported that the SBET is 58.86 m2.g−1. Regarding the SBET differed considerably when changing the material used in the nanocomposite’s preparation. Accordingly, it was found that the formation of PEDOT matrix increases the SBET of WO3−GO to 103.92 m2.g−1 and the corresponding Vpore to 0.11 cm3.g−1. Therefore, the polymer backbone in the PEDOT@WO3−GO structure would presumably provide additional space and volume for the diffusion of ions during the electrochemical charging and discharging processes.”
Point 7: Why the author only presents the capacitive behaviors of the PEDOT@WO3−GO hybrid electrodes in a 3−electrode system? The performance of the ASC devices or SSC device (2−electrode system) is missing? Asymmetric supercapacitors (ASCs) assembled using two dissimilar electrode materials offer a distinct advantage of wide operational voltage window, and thereby significantly enhance the energy density.
Response 7: Yes, that's right. The capacitive behaviors of the PEDOT@WO3−GO hybrid electrodes in the 2-electrode system provide the added advantage of a wide operational voltage window, and thereby significantly enhance the energy density. We will work in the future to apply this system.
Point 8: Fit the EIS plots in Figure 8b with a proper equivalent circuit diagram and discuss the corresponding solution resistance, charge transfer resistance, and Warburg impedance.
Response 8: We fitted the EIS diagrams in Figure 8-b with a suitable equivalent circuit diagram and discussed the corresponding solution resistance, charge transfer resistance, and Warburg resistance. (see lines 431-440)
“In order to interpret the data collected from EIS, Randles cell as an equivalent circuit proposed was utilized, shown in the inset Figure 8-b, where (Rs) is the resistance of the solution, (ZW) is the Warburg impedance for the diffusion of redox. (Cdl) is the double-layer capacitance, and (Rct) is the charge-transfer resistance. For the PEDOT@WO3−GO electrode, the Rct was about 9.7 Ω, which is less than that of WO3−GO material showing the Rct about 17.8 Ω. The ZW of the PEDOT@WO3−GO, which appeared in the low frequency region and corresponded to diffusion-controlled process, was highly less than that of WO3−GO, indicating the abundance of ions on electrode surface causing a decrease in impedance. In term of the Cdl, there was no difference between the two electrodes.”

Round 2
Reviewer 1 Report
The author provides a better explanation for the concerns raised. This manuscript can be published after minor revision.
1. The authors should pay more attention on the format of full-text such as superscript and subscript, avoiding abbreviation in title, abstract and keywords.
2. Suggest to remove the common characterization such as XRD, FTIR, TEM, TGA, XPS, TGA 25 and BET.
no
Author Response
Point 1: The authors should pay more attention on the format of full-text such as superscript and subscript, avoiding abbreviation in title, abstract and keywords.
Response 1: We paid more attention on the format of full-text, and we deleted all abbreviation in title, abstract and keywords. For example we put the following formula in the title:
“Poly(3,4-ethylenedioxythiophene)@Tungsten triox-ide−Graphene oxide”
Point 2: Suggest to remove the common characterization such as XRD, FTIR, TEM, TGA, XPS, TGA and BET.
Response 2: We removed the common characterization
“The structural and morphological analysis of prepared electrodes were extensively characterized by XRD, FTIR, TEM, TGA, XPS, TGA and BET.”

Reviewer 2 Report
Thank you for your careful consideration and revisions. I agree with most of your answers. However, I still have one question, which needs to be revised. Therefore, I'd like to ask authors to address the following issue prior to my full recommendation for acceptance to be published in “Nanomaterials”.
Question 1: The expressions and formats of the manuscript need to be checked carefully again. For example, “(a): GCD of PESOT@WO3-GO; (b): GCD of WO3-GO at various current densities and at a scan rate of 10 mv.s-1;” The caption in Figure 7 (a) and (b) are still confusing.
Author Response
Point 1: The expressions and formats of the manuscript need to be checked carefully again. For example, “(a): GCD of PESOT@WO3-GO; (b): GCD of WO3-GO at various current densities and at a scan rate of 10 mv.s-1;” The caption in Figure 7 (a) and (b) are still confusing.
Response 1: We re-examined the expressions and formats of the manuscript, and we corrected the typo in Figure 7:
“Figure 7. (a): GCD of PEDOT@WO3-GO; (b): GCD of WO3-GO at various current densities and at scan rate of 10 mv.s−1; (c): Specific capacitance curves of electrodes at different current densities; and (d): Cycling stability of electrodes: specific capacitance versus cycle number measured at a current density of 0.1 A·g−1.”

Reviewer 3 Report
Dear Authors,
Thanks for the answers...please specify one query: Which method (equations 1 and 2) did you use to calculate the capacity?
Thanks
Kind Regards
Aniruddha Mondal
Author Response
Point 1: Thanks for the answers.. please specify one query: Which method (equations 1 and 2) did you use to calculate the capacity?.
Response 1: We corrected the equation used to calculate the capacity (see equation 1):
|
(1) |

Reviewer 4 Report
The author has made adequate modification according to the comments, it can be accepted for publication in its current version.
No.
Author Response
Point 1: The author has made adequate modification according to the comments, it can be accepted for publication in its current version.
Response 1:
Dear Reviewer,
The authors would like to extend great thanks to you for your time and valuable comments and suggestions, which helped us improve our paper to a more scientific level.
Yours sincerely.
